# Morphometric Analysis of the Mandibular Canal and Its Anatomical Variants in a Chilean Subpopulation: Cone Beam Computed Tomography Study

**DOI:** 10.3390/diagnostics14171914

**Published:** 2024-08-30

**Authors:** Jacob Guzmán, Jaime Abarca, Pablo Navarro, Ivonne Garay, Josep Arnabat-Domínguez, Pablo Betancourt

**Affiliations:** 1Program of Master in Dental Science, Facultad de Odontología, Universidad de La Frontera, Temuco 4811230, Chile; j.guzman04@ufromail.cl; 2Faculty of Dentistry, Universidad San Sebastián, Sede Patagonia, Puerto Montt 5480000, Chile; jabarcareveco@gmail.com; 3Facultad de Ciencias de la Salud, Universidad Autónoma de Chile, Temuco 4810101, Chile; pablo.navarro@ufrontera.cl; 4Department of Integral Adultos, Facultad de Odontología, Universidad de La Frontera, Temuco 4780000, Chile; 5Private Practice, Radiologist, Temuco 4780000, Chile; ivogaray@gmail.com; 6Faculty of Medicine and Health Sciences, University of Barcelona, 08907 Barcelona, Spain; joseparnabat@ub.edu; 7Idibell Institute, 08908 Barcelona, Spain; 8Endodontic Laboratory, Center for Research in Dental Sciences (CICO), Faculty of Dentistry, Universidad de La Frontera, Temuco 4780000, Chile

**Keywords:** mandibular canal, inferior alveolar nerve, anterior loop, retromolar canal, CBCT

## Abstract

The inferior alveolar nerve (IAN), contained within the mandibular canal (MC), is a structure prone to damage in clinical and surgical procedures on the mandible. This study aimed to analyze the MC morphology and its anatomical variants in a Chilean subpopulation using cone beam computed tomography (CBCT). In total, 342 images from patients with the following parameters were observed: 120 kV, 9 mA, field of view 12 × 9 MC, and voxel size 0.12 mm. The average length of the MC recorded a mean value of 70.493 ± 4.987 mm on the right side and 70.805 ± 5.177 mm on the left side. The location of the mental foramen (MF) was most frequently found between the mandibular first and second premolar. The greatest bone thickness was found at the level of the basilar border of the 2MPM. The root closest to the MC was distal from the mandibular second molar. The lowest thickness was in the lingual area at 1MM. The prevalence of the anterior loop (AL) (61.5%) and the retromolar canal (RC) (17.5%) showed no significant differences between men and women. The results obtained showed that the morphology of the MC in the Chilean subpopulation can vary.

## 1. Introduction

The inferior alveolar nerve (IAN) is a terminal branch of the trigeminal nerve, which enters the mandibular canal (MC) through the mandibular foramen and emerges through the mental foramen to innervate the soft tissues. The MC is closely related to the roots of the mandibular posterior teeth. It is crucial to consider this relationship in clinical treatments across several dental disciplines and numerous surgical operations involving the mandible, such as dental implantology, maxillofacial surgery, and anesthesia [1,2,3,4].

It is widely recognized that environmental factors influence craniofacial growth [5]. The forces exerted by muscles are crucial to bone growth and development.

Additionally, research indicates that the consistency of a diet plays a crucial role in influencing craniofacial growth and development [6]. It has been observed that subjects on a hard diet show more pronounced sagittal and vertical development of the mandible compared to those on a soft diet [7]. Moreover, the consistency of a diet affects the morphology of the mandible, including changes in the condylar region, the mandibular angle, and the mandibular body [7].

In some individuals, the IAN extends anterior to the MF, reverses its direction, and forms a “loop” before emerging and innervating the surrounding tissues. The anterior loop (AL) can be identified by the existence of two separate canals or the elongation of the MC [8]. This anatomical variation becomes relevant during implant placement in the intraforaminal area. The MC may present a prevalent bifurcation in the retromolar area, the retromolar canal (RC). The presence of this bifurcation leads to complications such as hemorrhage or neurosensory disorders in procedures involving the retromolar trigone [9].

Considering the possible difficulties during mandible work, many methods have been suggested for morphometrically assessing the IAN and its course. Imaging studies such as periapical radiographs and panoramic radiographs have been used by several authors [10,11]. However, the use of these examinations is limited by the overlapping of structures, which distorts the morphological and morphometric analyses. It has been described that the use of cone beam computed tomography is more effective in detecting and analyzing mandibular structures than 2D studies [11,12,13].

Most of the studies that evaluated the morphology of the MC were conducted in European, African, Indian, and American populations. Therefore, there is a clear gap in current knowledge about the morphological characterization of the MC and its variations in the Chilean population.

In the present study, morphometric measurements of the MC, the proximity of the MC to the mandibular root apexes, the bone thicknesses surrounding the MC, and the prevalence of RC and AL in a population from southern Chile were analyzed using CBCT.

Considering all the abovementioned factors, this study aimed to obtain precise measurements of the MC and surrounding structures to establish a standardized metric pattern. This pattern could then serve as a useful tool for physicians in selecting the most suitable therapeutic method for each case while considering the anatomical variances that may be present in the MC.

## 2. Materials and Methods

A retrospective cross-sectional observational study was designed. The Scientific Ethics Committee of the Universidad de La Frontera approved the study, page 076/2022. A total of 342 images from the image bank of the Dental Teaching Assistance Clinic (*Clínica Odontológica Docente Asistencial*, CODA) of the Universidad de La Frontera, Temuco, Chile, taken from January 2016 to August 2023, were analyzed. An examiner (J.G) who received training from a specialist in dentomaxillofacial radiology (I.G) evaluated the samples.

The sample size was defined as follows:$$n=\frac{N\times Z_{\alpha }^{2}\times p\times q}{d^{2}\times \left(N-1\right)+Z_{\alpha }^{2}\times p\times q}$$
$$n=\frac{100\times 1.96^{2}\times 0.5\times 0.5}{0.05^{2}\times \left(100-1\right)+1.96^{2}\times 0.5\times 0.5}$$
n=79.51≈80
n=80 images
where

*N* = estimated population;

*Z_α_* = reliability;

*p*, *q* = estimated proportion;

*d* = estimated error.

When performing sampling adjustment, the following was obtained:$$n^{\prime }=\frac{n}{1+\frac{n}{N}}$$
$$n^{\prime }=\frac{100}{1+\frac{80}{100}}=55.55\approx 56$$
$$n^{\prime }=56\;\mathrm{images}$$

The minimum sample size to ensure results was 56 images.

The following inclusion criterion was applied: Men and women over 15 years of age with all their teeth and complete apical closure of the mandibular posteroinferior teeth were included. The excluded samples were from patients with severe periodontal pathologies or implants or undergoing any maxillofacial surgery on the mandible; patients with severe dental malpositions or facial asymmetries; and patients with supernumerary teeth in the posterior sector or with extruded teeth. In addition, distorted images and images that did not include the lower border of the mandible were excluded. Initially, to determine Cohen’s Kappa, an intraobserver concordance test was performed. Although it is more common to assess concordance in categorical data between observers, it can also be adapted to measure consistency in repeated assessments of the same observer. The Kappa index was calculated to see to what extent the same observer’s assessments are more consistent than expected by chance. The analysis was performed on 10% of the sample, and the correlation was accepted as agreement if it was higher than 0.8 (excellent). A sample of 200 CBTC images with an average age of 22.9 years ± 7.95 years were included in the study, divided into 49 men and 51 women.

The examinations were performed using a Pax Zenith CBCT unit (Vatech, Hwaseong-si, Republic of Korea) with the following parameters: 120 kV, 9 mA, field of view 12 × 9 MC, and voxel size 0.12 mm. The images were processed using the Ez 3D 2009 3D software (Vatech, Hwaseong-si, Republic of Korea) and displayed on a KDL-42W651A LED monitor (Sony, Minato, Japan).

Morphometric analyses of (i) MC length; (ii) MF location; (iii) distance of MC cortices to the dental apices of the first mandibular premolar (1MPM), second mandibular premolar (2MPM), first mandibular molar (1MM), and second mandibular molar (2MM); (iv) bone thicknesses surrounding the MC; and (v) the prevalence of RC and AL anatomical variations were performed. Multiplanar reconstruction (MPR) was used for the analysis, which allows axial (0.5 mm to 1 mm thick intervals), sagittal, and coronal sections to be viewed from the same series, making a flat or 2D image visible in 3D.

### 2.1. MC Length

The MC was traced in the sagittal section, which made it possible to see its trajectory. First, the entry of the IAN to the MC in the MF was identified in the coronal section, setting the first reference point here. The following points were placed equidistant from each other following the course of the MC. The last point was fixed on the MF, taking the curve of the AL into account (Figure 1). It was ensured that each point was well located on all CBCT sections. The length of the MC was provided by the software after tracing the MC, and the records were taken in mm.

### 2.2. Location of the MF

The location of the MF in relation to the mandibular teeth was classified as follows: (a) apical 1MPM, (b) between the root apex of 1MPM and 2MPM, (c) apical 2MPM, and (d) between the root apex of 2MPM and 1MM.

### 2.3. Distance of the Tooth Apices to the MC

The distance in mm between the upper edge of the MC to the apices of the posterior teeth was measured in the coronal sections. Prior to the measurement, the major axes of the tooth and its roots were fixed in axial, coronal, and sagittal sections. The teeth that were evaluated were 1MPM, 2MPM, 1MM, and 2MM. Due to the high anatomical variability of the third molar, it was excluded from this study.

### 2.4. Bone Thicknesses

To evaluate the bone thickness surrounding the MC, the major axes of the mandibular teeth were fixed, and three lines were drawn to three different points. Point A was fixed on the vestibular cortex, point B on the lingual cortex, and point C on the base of the mandible. Then, the average distances from these points to the MC were measured. After this recording, cortical thicknesses were measured to differentiate between medullary and cortical bone. Thicknesses were measured in relation to the major axis of the 2MPM, in the interradicular zone of the 1M, and in the interradicular zone of the 2M (Figure 2).

### 2.5. Anatomical Variants

The CBCT study permits longitudinal and oblique rotation of the sections, as well as buccolingual and posteroanterior movements. It was possible to view and analyze the anatomical variants, such as the AL and the RC, using these movements. Once the structures had been identified, their presence was confirmed in every section (Figure 3). The length of the AL was determined by counting the sequential coronal sections from the anterior edge of the MF to the disappearance of the AL, multiplied by the thickness of the section (0.5 mm) (Figure 4).

### 2.6. Statistical Analysis

A descriptive data analysis was performed, and the mean and its respective standard deviation were determined. The Kolmogorov–Smirnov normality test, *t*-test for independent samples, and *t*-test for related samples were performed.

The ICC statistic was used to assess the level of interobserver agreement. Pearson’s correlation coefficient was calculated to measure the intensity of the relationship between continuous variables. The chi-square test was employed to determine the association between categorical variables. The data analysis was performed with the IBM SPSS Statistics program (version 23.0). A *p* value < 0.05 was chosen as the significance threshold.

## 3. Results

In the intraexaminer agreement analysis, the ICC was 0.96 (excellent).

### 3.1. MC Length

The MC length on the right side recorded a mean value of 70.493 ± 4.987 mm; on the left, it was 70.805 ± 5.177 mm. The measurement ranges were 60.93 to 84.94 mm on the right side and 61.4 to 91.78 mm on the left side. Using the *t*-test for related samples, no significant differences were found when comparing the right and left sides (*p* = 0.170), but when comparing by sex, the length of the MC was significantly longer in men (*p* = 0.0001).

### 3.2. Location of the MF

Table 1 shows the position of the MF. The most common location was between the root apexes of the mandibular premolars.

### 3.3. Distance of the Root Apex to the MC

Two hundred teeth were analyzed: 1MPM, 2MPM, 1MM (mesial and distal root), and 2MM (mesial and distal root). The distances are shown in Table 2. In the case of the 1MPM, only 38 had a relationship that made it possible to measure the distance to the apex because the IAN emerges through the MF in the premolar area.

### 3.4. Mandibular Thicknesses

The thickest bone thickness was found at the level of the basilar border of the 2MPM. The lowest thickness was in the lingual area at 1MM. Details of the mandibular thickness measurements, differentiated between men and women, are reported in Table 3.

### 3.5. Retromolar Canal

The RC retromolar canal was found to have a prevalence of 17.5% of the total sample. On the right side, the RC was found to be 19%, while on the left, it was reported to be 23%. When using the chi-square test to compare the incidence of RC on both sides, the result showed a significance of *p* = 0.504. This indicates that there is no association between the two hemimandibles.

### 3.6. Anterior Loop

The AL was evaluated in 200 hemimandibles, where it was identified 124 times (61.5%). The mean and range of the AL was 2.860 mm and 0.0 to 5.5 mm, respectively. Using the *t*-test for independent samples, no statistically significant differences were found when comparing the left side (*p* = 0.521) or the right side (*p* = 0.998) between the two sexes. Table 4 summarizes the findings on the incidence and length of the anterior loop.

## 4. Discussion

CBCT is considered an important tool in the assessment of craniofacial anatomy and essential details regarding the thickness, dimensions, and positioning of the mandibular canal (MC) in relation to surrounding structures [13]. The advantages of CBCT include uniform magnification, three-dimensional reconstructions, precise geometric accuracy, reduced radiation exposure, and comparatively low cost [14]. It is widely recognized that CBCT significantly outperforms panoramic radiography in terms of both specificity and sensitivity [15]. The methodology of our research is grounded in previously published studies [8,16,17] where the visualization protocols for studying the morphometric characteristics of the MC and its anatomical variations have been validated. By employing a methodology consistent with these studies, we ensure the comparability of our results.

### 4.1. MC Length

The average MC length in our study was 70.64 mm. Consistent with previous research, the length of the MC in the left and right hemimandibles was comparable in populations in Chile and India [16,18]. However, statistically significant differences were found in the average length of the MC according to sex, being greater in men (*p* = 0.0001). Regarding the MC trajectory, a downward trajectory from the 2MM toward the 1MM was noted, followed by a slight upward trajectory toward the 2PMM in its posteroanterior trajectory, which aligns with prior findings [17,19]. Regarding the buccolingual relationship, it has been reported in the literature that the MC follows a course closer to the lingual cortex in the molar area and closer to the buccal cortex in the premolar area [4,18,20], with results similar to those reported in our study.

### 4.2. Location of the MF

It has been observed that the location of the MF varies according to the population studied [19]. Our investigation found that the MF was positioned between the premolars in 62% of the sites and in proximity to the apex of the 2MPM in 25.5% of the cases. In a study on a Spanish population [21], the MF was found between mandibular premolars in 56%, followed by under the 2MPM in 37%. Similar values were found by Neiva et al. (2004) in a Caucasian population; they reported 58% of cases with the MF between the premolars, followed by 42% with the MF under the 2MPM [22]. African populations exhibit a pattern of more distally located MF than European populations. In Malawian and Tanzanian populations, studies using skeletal samples described the tendency of the MF being located under the 2MPM followed by between the 2MPM and 1MM [23,24].

### 4.3. Distance from Root Apices to the MC

The proximity of the MC to the dental roots is likely the most clinically relevant information, given that a significant number of procedures can cause damage to the IAN. To accurately assess the interaction between the dental apices and the MC, teeth lacking occlusal contact were excluded, as they provide a risk of dental extrusion. Additionally, malpositioned teeth were also excluded. As in previous studies [2,25], the present study noted greater distances in men than women, as was the case in the left 2MPM (*p* = 0.43), the mesial and distal root of the left 2MM (*p* = 0.03 and *p* = 0.045, respectively), and in the mesial and distal root of the right 2MM (*p* = 0.006 and *p* = 0.013, respectively). This would indicate a higher percentage of risk of damage to the IAN in women, which agrees with studies carried out in populations from Iran, India, and Romania [4,18,20]. At the mesial root level of the 1MM, the MC tends to move away from the apices and then approach the 2MPM. This serpentine trajectory is consistent with reports by other authors [2,16,26].

Concerning the root apex closest to the MC, our results align with the bulk of the research included in this study, with the 2MM being closest to the MC. The results of this study resemble the measurements reported by Kovisto et al. in 2016 (1.51 mm) and Bruklein et al. in 2015 (2.3 mm) in the age group of our study; however, they vary significantly from those reported by Muñoz et al. in 2017 (3.9 mm) and Komal et al. in 2020 (3.2 mm) [2,16,18,26]. This can be attributed to the omission of certain criteria in these studies, such as dental extrusion in individuals with partial dentures or dental malposition. In addition, it has been observed that there is a correlation between increasing age and increasing distance between the MC and the root apexes [19], which could explain the differences in the results.

Researchers have reported a direct relationship between the MC and root apex. Our results showed a 41.5% direct relationship of the 2MM in some of its roots. When analyzing patients individually, it was found that in 24.5% of cases, the 2MM, on the right or left side, showed a direct communication with the MC. Burklein et al. (2015) and Aksoy U et al. (2018) reported a direct relationship among 15% and 16% of patients, respectively [2,25]. This observation may explain the higher incidence of damage to the IAN in procedures involving the 2MM [26].

### 4.4. Mandibular Thicknesses

In the Chilean population studied here, bone thickness between the MC and the basilar border decreased in the anteroposterior direction, coinciding with the findings reported in Sudanese [19] and Asian populations [27]. A comparison of the left and right sides yielded significant differences only in the thickness of the trabecular bone (*p* = 0.047) and cortical bone (*p* = 0.032) of the 2MM when the *t*-test for related samples was applied. There was greater thickness on the right side at both points, the only point with a significant difference. This differs from Kawashima et al. (2016), who reported higher values of vestibular bone thickness in the left hemimandibles in a US population [28]. When analyzing the differences by sex, it was noted that the thickness values in men were higher, which agrees with the reports by Ahmed AA et al. (2021) and Khorshidi et al. (2017) [4,19]. However, this result differs from that reported by Arias et al. (2020) and Kawashima et al. (2016), who found no significant differences between men and women [17,28]. The lower bone thickness in the two sexes was related to the lingual cortex of the 1MM, as described by Kovisto et al. (2016), Muñoz et al. (2017), Arias et al. (2020), and Khorshidi et al. (2017) [4,13,16,17]. In contrast, the greatest bone thickness was recorded at the level of the basilar border of the 2MPM.

### 4.5. Retromolar Canal

Like the MC, the RC has been studied using different imaging techniques, with different prevalences depending on the method. A systematic review carried out by Valenzuela-Fuenzalida et al. (2021) reported that the presence of the RC accounted for 33.6% of the total number of mandibles included in the different studies analyzed (4.577 mandibles) with a range of incidences from 8.8 to 75% with the different imaging techniques [29]. The differences in prevalence between panoramic radiographs and CBCT can be explained by the possibility of observing the mandible in 3D using the latter technique since this structure is difficult to detect with a two-dimensional radiograph [29,30]. The results of the present study showed a prevalence of 17.5% using CBCT, similar to the 24.5% reported in the Brazilian population [9], 26% in the Japanese population [31], and 22% in the Iranian population [32].

### 4.6. Anterior Loop

Different methodologies have been proposed for detecting the AL by CBCT, but there is still no consensus on the most correct method for its measurement. CBCT has proven to be a reliable tool for determining the anterior extension of the nerve, with average discrepancies of only 0.06 mm compared to anatomical measurements in cadavers [33]. The present study found the AL in 61.5% of patients, results similar to those reported by other authors, at 66.01% [27], 71% [33], and 75.6% [34]. On the other hand, prevalences of 40.2% have been described in panoramic radiographs [35].

Studies have reported an average AL length ranging from 0.4 mm to 6 mm. Apostolakis et al. (2012) reported an average length of 0.89 mm, and Raju et al. (2019) reported 1.63 mm [11,12]. Our results averaged 2.8 mm in length. Apostokalis et al. (2012) indicated that the longest AL in their study was 5.7 mm, similar to our study, which measured 5.5 mm [8]. It is worth noting that the longest AL reported in the literature is 11 mm [22].

One of the problems when measuring the AL is the ability to differentiate it from the incisive canal (IC). Mardinger et al. (2000) concluded that the diameter ranges from 0.5 to 2 mm [36]. However, some studies have reported a diameter as large as 6.6 mm [33,37]. This is why we made the differentiation: It has been shown that AL lengths greater than 2 mm are clinically relevant. Notably, 36% of the patients in this study had a length greater than 2 mm, similar to the 35.6% reported by Al-Siweedi SYA et al. (2014) in the Sudanese population [27].

The safety margin determined is 3 mm from the anterior edge of the MF. In 21.5% of the cases in our study, an AL greater than 3mm was recorded. Without the use of CBCT, a margin of 6 mm is considered safe, but it should be noted that an AL greater than 6 mm has been reported. This is why we think that the use of a CBCT study prior to the placement of implants in the intraforaminal area is mandatory.

## 5. Conclusions

The morphometric measurements observed in a Chilean MC subpopulation are similar on both sides of the mandible, although there may be sex-related differences. On average, the MC measures 70.4 mm on the right side and 70.8 mm on the left side. Women tend to have significantly less distance between the MC and the root apexes and often show a direct communication between the root apex and the MC. The AL was identified in 61.5% of the cases, while the RC was identified in 17.5% of the sites. The most frequent location of the MF was between the premolars (62%). The MC morphometric measurements detailed in this article can serve as a guide for clinicians, but the fact that the anatomy varies according to each individual must be considered.

## Figures and Tables

**Figure 1 diagnostics-14-01914-f001:**
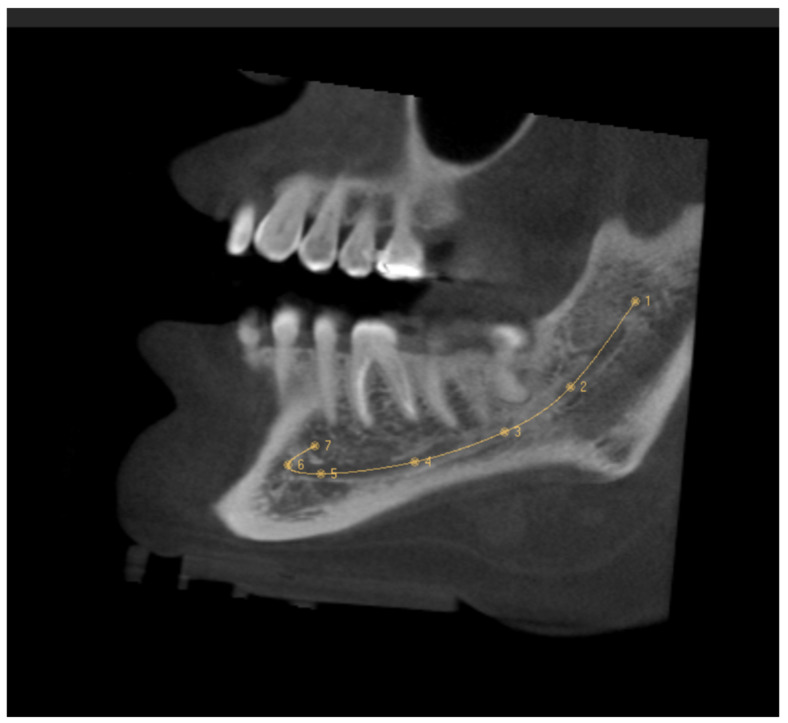
Sagittal slice CBCT image showing the tracing of the mandibular canal. The first reference point is located in the mandibular foramen, and the seventh reference point is in the mental foramen.

**Figure 2 diagnostics-14-01914-f002:**
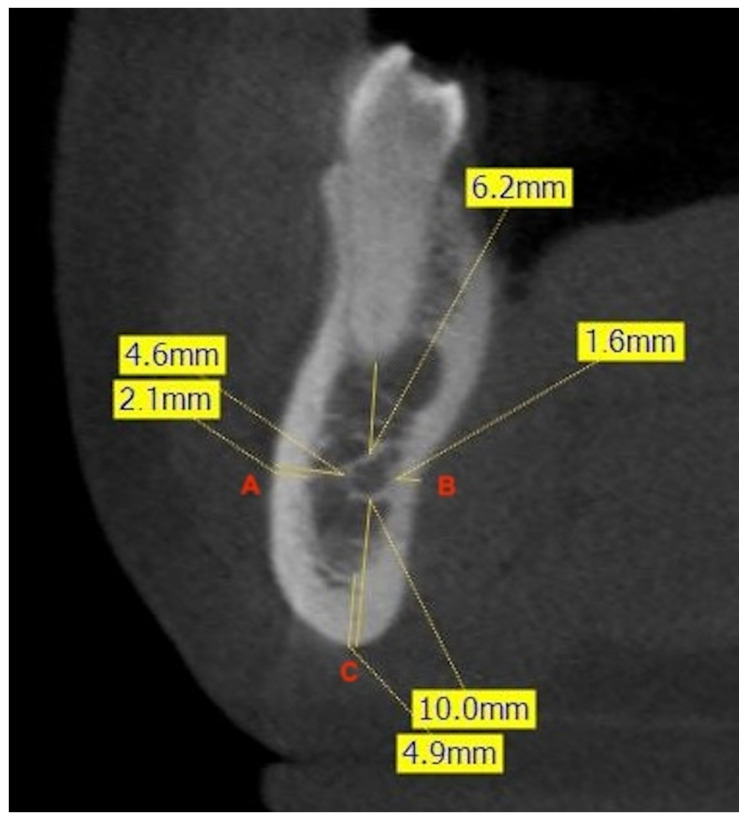
Example of measurement of the shortest linear distances (mm) at the level of the right second premolar. The frontal CBCT image shows the distance from the most buccal (A) and lingual (B) aspects of the canal to the corresponding cortical plates of the mandible, and the minimum linear distance between the inferior aspect of the canal and the inferior border of the mandible (C). The measurement of the thickness of the cortical bone at each point is also illustrated.

**Figure 3 diagnostics-14-01914-f003:**
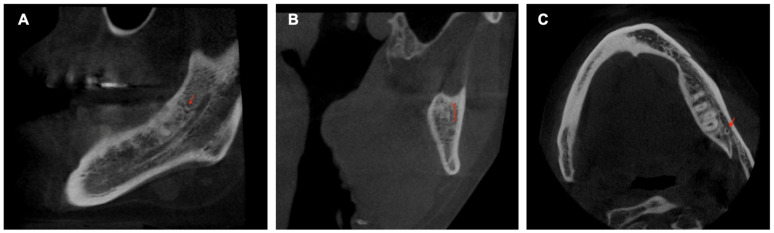
Sagittal (**A**), frontal (**B**), and axial (**C**) CBCT slides. The red arrow indicates the location of the retromolar canal.

**Figure 4 diagnostics-14-01914-f004:**
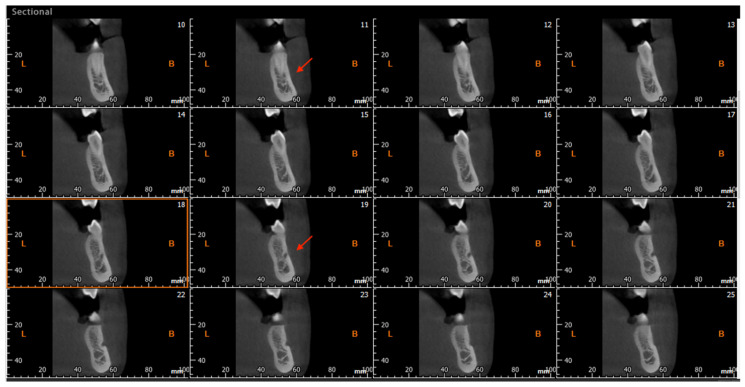
Cross-sectional CBCT reconstruction. The anterior loop is visible in images No. 11–19 (red arrows). The length was measured as 9 × 0.5 = 4.5 mm. In image No.18 (highlighted with an orange square), the junction of the loop with the mandibular canal is observed, making the canal appear ‘wider’.

**Table 1 diagnostics-14-01914-t001:** Frequency analysis of the location of mental foramen.

Location	Frequency	Percentage (%)
Apical to the first premolar	19	9.5
Between the premolars	124	62
Apical to the second premolar	51	25.5
Between the second premolar and the mesial root of the first molar	6	3
Total	200	100

**Table 2 diagnostics-14-01914-t002:** Distance of the mandibular canal to the root apex (mm).

Distance to the Root Apex (mm)	n	Average Distance to the Apex (mm)	SD	Right	n	Distance to the Left Apex (mm)	SD	Left	n	Distance to the Right Apex (mm)	SD
2nd Molar											
2MM D (Average)	200	1.61	1.86	4.7 R (Average)	100	1.5	1.71	3.7 D (Average)	100	1.73	2.01
2MM D (Man)	98	2.04	1.96	4.7 D (Man)	49	1.93	1.85	3.7 D (Man)	49	2.14	2.07
2MM D (Woman)	102	1.21	1.68	4.7 D (Woman)	51	1.33	1.88	3.7 D (Woman)	51	1.08	1.48
2MM M (Average)	200	1.87	2.01	4.7 M (Average)	100	1.95	2.04	3.7 M (Average)	100	1.79	1.98
2MM M (Man)	98	2.47	2.17	4.7 M (Man)	49	2.52	2.28	3.7 M (Man)	49	2.43	1.18
2MM M (Woman)	102	1.29	1.65	4.7 M (Woman)	51	1.4	1.61	3.7 M (Woman)	51	1.18	1.69
1st Molar											
1MM D (Totaled)	200	3.14	2.11	4.6 D (Average)	100	3.21	2.11	3.6 D (Average)	100	3.07	2.11
1MM D (Man)	98	3.5	2.31	4.6 D (Man)	49	3.61	2.3	3.6 D (Man)	49	3.39	2.33
1MM D (Woman)	102	2.8	1.85	4.6 D (Woman)	51	2.82	1.86	3.6 D (Woman)	51	2.77	1.85
1MM M (Totaled)	200	3.76	2.14	4.6 M (Average)	100	3.78	2.1	3.6 M (Average)	100	3.75	2.18
1MM M (Man)	98	4.08	2.21	4.6 M (Man)	49	4.04	2.2	3.6 M (Man)	49	4.12	2.23
1MM M (Woman)	102	3.46	2.03	4.6 M (Woman)	51	3.53	1.98	3.6 M (Woman)	51	3.38	2.09
2nd Premolar											
2MPM (Totaled)	199	3.44	2.38	4.5 (Average)	100	3.57	2.35	3.5 (Average)	99	3.3	2.41
2MPM (Man)	97	3.88	2.48	4.5 (Man)	49	3.91	2.46	3.5 (Man)	48	3.85	2.52
2MPM (Woman)	102	3.02	2.21	4.5 (Woman)	51	3.25	2.21	3.5 (Woman)	51	2.79	2.2
1st Premolar											
1MPM (Totaled)	38	3.89	1.5	4.4 (Average)	18	4.02	1.59	3.4 (Average)	20	3.79	1.44
1MPM (Man)	16	4.22	1.53	4.4 (Man)	8	3.88	4.13	3.4 (Man)	8	4.56	1.51
1MPM (Woman)	22	3.66	1.46	4.4 (Woman)	10	4.13	1.68	3.4 (Woman)	12	3.27	1.18

**Table 3 diagnostics-14-01914-t003:** Cortical and trabecular jawbone thicknesses (mm).

Average Thickness (mm)	Men						Women					
	Total Thickness	SD	Min	Max	Average Cortical Bone	SD Cortical Bone	Total Thickness	SD	Min	Max	Average Cortical Bone	SD Cortical Bone
2MPM												
Buccal	3.35	1.74	0	7	2	1.07	3.3	1.83	0	6.5	1.74	0.84
Lingual	2.98	1.42	0.5	7.7	1.86	0.42	2.98	1.48	0	9.2	1.77	0.37
Basilar B.	8.59	1.87	5.5	15.9	3.67	0.68	7.63	1.98	3.5	13.1	3.62	0.62
1MM												
Buccal	5.37	1.15	2.5	8.2	2.17	0.43	5.45	1.2	1.7	9	2.18	0.32
Lingual	1.54	0.74	0.5	4.4	1.39	0.48	1.7	0.75	0.4	4.4	1.44	0.41
Basilar B.	7.43	1.74	4	13.2	3.63	0.81	6.5	1.31	3.9	10	3.49	0.5
2MM												
Buccal	6.17	1.63	3.1	9.8	2.34	0.47	5.99	1.17	3.1	9	2.48	0.41
Lingual	1.55	0.63	0.5	3.6	1.42	0.46	1.65	0.578	0.5	3.2	1.52	0.43
Basilar B.	6.95	1.96	3.3	13.6	3.35	0.64	6.27	1.52	3	11.2	3.19	0.52

**Table 4 diagnostics-14-01914-t004:** Frequency and average length of the anterior loop.

	Frequency	Women	Men	Average Length (mm)	Standard Deviation (mm)
Left anterior loop	69	34	35	2.7391	1.34932
Right anterior loop	54	27	27	2.9811	1.23242

## Data Availability

No new data were created or analyzed in this study. Data sharing is not applicable to this article.

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
