# Peer review of "Morphometric Analysis of the Mandibular Canal and Its Anatomical Variants in a Chilean Subpopulation: Cone Beam Computed Tomography Study"

_diagnostics, 2024, doi:10.3390/diagnostics14171914_

Round 1

Reviewer 1 Report

Comments and Suggestions for Authors

In the beginning of the methodology, it would be good to mention the type of the study design, my understanding it is a retrospective crossectional study

In the 2nd paragraph, although it is mentioned that the kappa analysis was done, but how was the caliberation performed, was inter or intra examinar? this needs elaboration in the methods section

the discussion should start with discussing the methodology which validates the methodology undertaken rather than straightaway comparing the results to previous studies

it will be good to include the sample size calculation to determine the minimum sample needed, to ensure the results can be considered valid

Author Response

Dr. Andreas Kjaer

Editor-in-Chief

Diagnostics

Dear Dr. Kjaer,

We would like to express our sincere gratitude to you and the reviewers for the thorough evaluation and insightful comments provided on our manuscript [ID: diagnostics-3152361 / “Morphometric analysis of the mandibular canal and its anatomical variants in a Chilean subpopulation. Cone beam computed tomography study”]. We greatly appreciate the time and effort invested in reviewing our work.

The comments and suggestions offered by the reviewers have led to enriching discussions among our team, ultimately enhancing the quality of our manuscript. We believe that the revisions made in response to these valuable inputs have significantly strengthened the initial version of our submission.

Please find below our detailed responses to each of the reviewers' comments. We hope that the revised manuscript meets the high standards of Diagnostics.

Once again, thank you for the opportunity to improve our work through this constructive review process.

Sincerely,

Dr. Pablo Betancourt, PhD

Associate Professor
Research Centre in Dental Sciences (CICO) Endodontic Laboratory, Faculty of Dentistry Universidad de La Frontera
Orcid 0000-0002-9903-2920

Comments and Suggestions for Authors Reviewer 1

1. In the beginning of the methodology, it would be good to mention the type of the study design, my understanding it is a retrospective crossectional study.

Rp. Thank you for your valuable suggestion. We agree that including the study design type strengthens the methodological clarity of the manuscript. Accordingly, we have added the study design in line 79: "A retrospective cross-sectional observational descriptive study was designed."

This addition should provide clearer context for readers regarding the nature of the study.

2. In the 2nd paragraph, although it is mentioned that the kappa analysis was done, but how was the caliberation performed, was inter or intra examinar? this needs elaboration in the methods section

Rp. We appreciate your insightful comment. The requested explanation regarding the Kappa analysis has been added to the manuscript. You can find this information in lines 115-119: “Initially Cohen's Kappa, intraobserver concordance test was performed, although more common to assess concordance in categorical data between observers, it can also be adapted to measure consistency in repeated assessments of the same observer. The Kappa index was calculated to see how much more of the same observer’s assessments are more consistent than expected by chance”.

3. The discussion should start with discussing the methodology which validates the methodology undertaken rather than straightaway comparing the results to previous studies

Rp. Thank you for your valuable suggestion. We agree that starting the discussion by addressing the methodology used can provide a stronger foundation for the subsequent discussion. We have incorporated this suggestion, and you can see the changes in lines 252- 261 of the manuscript. We believe that the added paragraph enhances the manuscript by providing a more comprehensive discussion of the methodology before moving on to compare our results with previous studies.

4. It will be good to include the sample size calculation to determine the minimum sample needed, to ensure the results can be considered valid.

Rp. We completely agree that this is an important aspect to clarify in the methodology section. We have decided to include the statistical formula used to calculate the minimum required sample size, as requested by Reviewer 1. You can find this addition between lines 88 and 107. We believe that this inclusion strengthens the methodological rigor of the manuscript. Thank you for the insightful comment.

Reviewer 2 Report

Comments and Suggestions for Authors

Dear Authors,

Thank you for submitting your research to the journal. I found your research really interesting and well executed in general. Please find below my suggestions which I think will improve the presentation of your research. 

Introduction : Your introduction is really short. I would advise you to add more information about previous studies. It will be valuable to add more information about morphometrics and how this technique helped us to achieve certain outcomes. For example the effect of diet consistency on mandibular morphology. Here are some suggested references:
-Tsolakis, I.A.; Verikokos, C.; Perrea, D.; Bitsanis, E.; Tsolakis, A.I. Effects of diet consistency on mandibular growth. A review. J. Hell. Vet. Med. Soc. 201970, 1603–1610.                                    

-Tsolakis, I.A.; Verikokos, C.; Perrea, D.; Alexiou, K.; Gizani, S.; Tsolakis, A.I. Effect of Diet Consistency on Rat Mandibular Growth: A Geometric Morphometric and Linear Cephalometric Study. Biology 202211, 901

Materials and methods: The inclusion and exclusion criteria are well written  I would advise you to include the final sample number in this section and not in the results section. Furthermore you should keep consistent your sample. In the materials and methods you write about images in the results about patients. Statistical analysis is well presented.

Results: well presented other than including the total number of patients in the materials and methods section there is no need to make further changes

Conclusion: well written 

Author Response

Dr. Andreas Kjaer

Editor-in-Chief

Diagnostics

Dear Dr. Kjaer,

We would like to express our sincere gratitude to you and the reviewers for the thorough evaluation and insightful comments provided on our manuscript [ID: diagnostics-3152361 / “Morphometric analysis of the mandibular canal and its anatomical variants in a Chilean subpopulation. Cone beam computed tomography study”]. We greatly appreciate the time and effort invested in reviewing our work.

The comments and suggestions offered by the reviewers have led to enriching discussions among our team, ultimately enhancing the quality of our manuscript. We believe that the revisions made in response to these valuable inputs have significantly strengthened the initial version of our submission.

Please find below our detailed responses to each of the reviewers' comments. We hope that the revised manuscript meets the high standards of Diagnostics.

Once again, thank you for the opportunity to improve our work through this constructive review process.

Sincerely,

Dr. Pablo Betancourt, PhD

Associate Professor
Research Centre in Dental Sciences (CICO) Endodontic Laboratory, Faculty of Dentistry Universidad de La Frontera
Orcid 0000-0002-9903-2920

Comments and Suggestions for Authors Reviewer 2

Dear Authors,

Thank you for submitting your research to the journal. I found your research really interesting and well executed in general. Please find below my suggestions which I think will improve the presentation of your research.

1. Introduction : Your introduction is really short. I would advise you to add more information about previous studies. It will be valuable to add more information about morphometrics and how this technique helped us to achieve certain outcomes. For example the effect of diet consistency on mandibular morphology. Here are some suggested references:

J. Hell. Vet. Med. Soc. 2019, 70

-Tsolakis, I.A.; Verikokos, C.; Perrea, D.; Bitsanis, E.; Tsolakis, A.I. Effects of diet

consistency on mandibular growth. A review.

1610.

, 1603–

-Tsolakis, I.A.; Verikokos, C.; Perrea, D.; Alexiou, K.; Gizani, S.; Tsolakis, A.I. Effect of Diet Consistency on Rat Mandibular Growth: A Geometric Morphometric and Linear Cephalometric

Study.

Biology 2022, 11, 901

Rp. Dear Reviewer, we appreciate your comment and agree that this addition can strengthen the introduction section. We have included the point you mentioned regarding the relationship between diet and mandibular morphology, which is indeed an interesting and underexplored topic. We have also added new literature to delve deeper into this subject. You can find the newly added paragraph

between lines 43 and 50.

2. Materials and methods: The inclusion and exclusion criteria are well written. I would advise you to include the final sample number in this section and not in the results section. Furthermore you should keep consistent your sample. In the materials and methods you write about images in the results about patients. Statistical analysis is well presented.

Rp. We have made the suggested changes, moving the final sample size from the results section to the materials and methods section. We agree with your observation regarding the need for consistency in terminology and have standardized the reference to "images" throughout the manuscript, rather than "patients." We appreciate your comment, as we believe it strengthens the paper. The phrase " A sample of 200 CBCT images with an average age of 22.9 years +/- 7.95 years were included in the study, divided into 49 men and 51 women " has been relocated to the materials and methods section, and we have maintained consistency by referring to the sample as 200 CBCT images.

3. Results: well presented other than including the total number of patients in the materials and methods section there is no need to make further changes

Rp. Following your suggestion, we have included the total number of images in the materials and methods section. No further changes were made to the results section, as it is now aligned with the updated content. We appreciate your input.

4. Conclusion: well written

Rp. Thank you so much

Round 2

Reviewer 2 Report

Comments and Suggestions for Authors

Dear Authors,

thank you for following my suggestions. According to my opinion your paper is good to be published